# Effects of Genotype and Climate on Productive Performance of High Oleic *Carthamus tinctorius* L. under Rainfed Conditions in a Semi-Arid Environment of Sicily (Italy)

**DOI:** 10.3390/plants12091733

**Published:** 2023-04-22

**Authors:** Mario Licata, Davide Farruggia, Nicolò Iacuzzi, Roberto Matteo, Teresa Tuttolomondo, Giuseppe Di Miceli

**Affiliations:** 1Department of Agricultural, Food and Forest Sciences, Università degli Studi di Palermo, Viale delle Scienze 13, 90128 Palermo, Italy; mario.licata@unipa.it (M.L.); davide.farruggia@unipa.it (D.F.); teresa.tuttolomondo@unipa.it (T.T.); giuseppe.dimiceli@unipa.it (G.D.M.); 2Research Centre for Cereal and Industrial Crops, Consiglio per la Ricerca in Agricoltura e l’Analisi dell’Economia Agraria, Via di Corticella 133, 40128 Bologna, Italy; roberto.matteo@crea.gov.it

**Keywords:** high oleic acid safflower, genotype, environment, fatty acids, crop residues

## Abstract

Safflower (*Carthamus tinctorius* L.) is a promising oilseed crop for cultivation in central Southern Italy due to its high tolerance to drought and salinity stress and appreciable seed and oil yields. The genetic diversity of cultivars and climate factors can affect fatty acid composition and yield traits. This study aimed to assess the effects of genotype and climate conditions on the productive performance of eight high oleic safflower genotypes under rainfed conditions in Sicily (Italy). These genotypes were compared to the Montola 2000 cultivar, which was used as a reference. Tests were carried out during the growing seasons of 2014–2015 and 2015–2016. The experimental design was a randomized complete block design with three replications. Morphological and yield components were significantly affected by genotype while the year had a significant effect on plant height only. In general, CTI 17 produced the highest seed yield (1.40 t ha^−1^) and oil yield (0.58 t ha^−1^). The seed oil content was found on 40.2% of dry matter, on average. The “genotype” factor significantly affected oil content and fatty acid composition. Oleic acid content was on average 66.1% and did not vary greatly over the two growing seasons. The above- and belowground plant parts showed the highest carbon content and the lowest nitrogen content as a percentage of dry matter. The results indicate that, under rainfed conditions, yield parameters of high oleic safflower genotypes can be profitable in southern Italy though significantly dependent upon genotype.

## 1. Introduction

In recent years, attention to climate change has assumed increasingly greater importance in agriculture due to the effects on crop production. Emerging evidence highlights that climate change significantly impacts agricultural areas and causes manifest yield loss for traditional crops as a result of more unpredictable weather conditions [1,2,3,4]. A promising solution to this problem in agriculture is provided by crop diversification through the use of underutilized, minor and neglected crops [5,6,7]. The introduction of these crops into traditional crop rotations would increase agro-biodiversity and buffer against crop vulnerability to climate change, pests and diseases. It would also provide various food sources to address both food and nutritional safety, as described by Mustafa et al. [2].

Focusing on the Mediterranean region, as one of the most developed cereal systems in the world, recent studies report that climate change seems to induce severe yield losses in traditional cereals, mainly due to an increase in average spring temperatures [8,9]. In this region, crop diversification could be introduced to reduce the impact of climate change on traditional crops. Underutilized oilseed crops in rotation with fall/winter cereals, for example, can provide a series of agronomic and economic benefits to farmers, including reduced agricultural inputs, limited abiotic and biotic stresses, and improved income due to their evident potential [7,10,11,12]. Furthermore, the choice of oilseed crops can also be profitable due to the fact they contribute to the production of renewable energy directly on farms and permit a reduction of dependence on fossil fuel energy sources, in accordance with the existing European renewable energy directives [13].

Of the underutilized/minor oilseed crops, safflower (*Carthamus tinctorius* L.) can be successfully introduced into rainfed cereal-based cropping systems in southern Mediterranean areas due to its tolerance to cold, drought and soil salinity, and its reduced needs for agricultural inputs [7,14]. This crop is largely known for oil production due to its high nutritional properties [15,16,17]. The quality of the oil is determined by the fatty acid composition, which is a combination of saturated fatty acids (SFA) and unsaturated fatty acids (UFA) [18]. Linoleic and oleic acids are the most abundant of the unsaturated acids and represent 90% of the total fatty acid content. The remaining 10% corresponds to saturated fatty acids, such as palmitic and stearic acids [19,20]. Based on the oil composition of the seed, safflower varieties are grouped into two types, one is characterized by high levels of linoleic acid and the other is rich in oleic acid. Traditional safflower oil, rich in polyunsaturated linoleic acid, is valued for health reasons as the high linoleic content leads to a significant reduction in cholesterol levels in human blood [21]. However, it is not suitable for prolonged frying due to low oxidative stability at high temperatures. In contrast, oil which is rich in monounsaturated oleic acid shows high oxidative stability, making it suitable for cooking and an alternative to olive oil in arid and semi-arid regions of the world [22]. Furthermore, oleic acid is characterized by high single-point unsaturation, a characteristic which is highly valued by the industry. Safflower oil which is rich in oleic acid can be converted into varnishes, alcohols, paints, lubrificants, cosmetics, detergents and bio-based plastics [21]. It also shows significant potential for use in biodiesel production, as reported by Valcir Kniphoff de Oliveira et al. [23].

Previous research has demonstrated that the linoleic and oleic content in the oil is mainly affected by genetic aspects [22,24,25,26] and agronomic practices such as fertilization, irrigation, harvest time and sowing date [27,28,29,30,31,32]. However, environmental factors (air temperature and moisture levels in particular) during seed maturation greatly affect fatty acid synthesis and the relative content of linoleic and oleic acids in the seed oil [22,33]. Concerning oilseed crops belonging to the *Asteraceae* family, such as sunflower (*Helianthus annuus* L.), it is well known that at higher minimum temperatures an increase in oleic acid and a corresponding decrease in linoleic acid could be expected in the oil content during seed maturation [34]. Regarding safflower, little information is available in the literature [35,36] and the effect of climate conditions on the oleic acid content of high oleic acid varieties has not been well investigated. Furthermore, an evaluation of the above- and belowground crop residue production, as influenced by genotype and climate conditions, represents a topic of great interest in relation to the different uses of biomass.

In Southern Italy, very limited research on safflower varieties has been carried out recently [25] for a variety of reasons. Farmers are reluctant to include minor or underutilized crops in rotation with fall/winter cereals or annual legumes. There is no local market and locally adapted varieties are not available. There has been greater importance given to other oilseed crops. However, this area is highly suited to safflower cultivation and production due to favourable soil and climate characteristics. This species could be easily suggested to farmers for diversification of crop production due to its potential application in the food industry, as well as others.

This study was aimed at assessing the effect of genotype and climate conditions on fatty acid composition, oil content, oil yield and crop residues of eight high oleic safflower genotypes under rainfed conditions.

## 2. Results

### 2.1. Analysis of Rainfall and Temperature in the Study Area

Rainfall and temperature trends during the 2014–2015 and 2015–2016 growing seasons are shown in Figure 1.

Rainfall was different in the two growing seasons. In particular, the total rainfall that occurred during the first growing season was 780 mm, reasonably different from that of the second growing season (518 mm) and the long-term average (600 mm) of the study area, with prevalent distribution in the months of December and March. The heaviest rainfall was recorded in the second 10-day period of February 2015 (91 mm). In the two growing seasons, total monthly rainfall levels showed a different trend. Rainfall levels were well distributed throughout the 2014–2015 growing season and this permitted an increase in soil water availability for a longer period compared to the 2015–2016 growing season. This affected crop vegetative and reproductive phases in the two growing seasons.

Regarding temperature, average minimum and maximum temperatures in the two growing seasons were similar and consistent with the ten-year average temperature (18.1 °C). Temperatures decreased progressively from November to February and then increased up to July/August, when ripening occurred. The highest maximum temperature (35.9 °C) was recorded in the third 10-day period of July 2015, whilst the lowest minimum temperature (4.9 °C) was logged in the first 10-day period of February 2015. During winter, the plants did not show any frost damage despite minimum temperatures falling below 6 °C in each growing season. Furthermore, in summer, any heat damage was recorded in the plants when maximum daily temperatures rose above 30 °C and rainfall was absent, confirming the high drought tolerance of safflower.

### 2.2. Plant Growth Stages

In the two years of study, the length of the growth cycle differed among the safflower genotypes (Table 1).

In the first growing season, the crop had a shorter growth cycle compared to the second growing season, with an average cycle length of 189 days from sowing to harvest. This was probably due to the different distribution of rainfall during spring. In fact, in the first growing season, the poor distribution of rainfall in April and May together with increasing average air temperatures determined evident stress conditions that led to early maturity in safflower genotypes. In each year, no great differences were found among the genotypes. When comparing the two years, the earliest genotype was Montola 2000 (186 days, on average) and the later-maturing accession was CTI 13 (192 days, on average). The germination stage occurred within 8 to 12 days in 2014–2015 and within 11 to 14 days in the 2015–2016 growing seasons depending on minimum and maximum air temperature. At the end of winter, when the maximum temperature increased, plants grew fast and the internodes became evident. On average, stem elongation occurred within 69 days from the sowing date during the test period; however, a different range was apparent, in terms of length, when comparing the 2014–2015 and 2015–2016 growing seasons. With regard to the flowering stage, on average this stage occurred earlier in the 2014–2015 growing season. The safflower genotypes, however, showed little differences between the growing seasons. It is important to highlight that rainfall levels and air temperature affected the length of this stage. When comparing the nine genotypes, Montola 2000, CTI 5 and CTI 10 reached the fruit ripening stage earlier than others in the two growing seasons. This phenological stage occurred, on average, within 175 days in the 2014–2015 and 182 days in the 2015–2016 growing season. The senescence stage occurred when maximum air temperatures rose to 25/30 °C and rainfall was absent. Plants became dry, the seeds were hard and white and the capitula were brown. Finally, the safflower genotypes showed a growth cycle ending before the first 10-day period of July in both the growing seasons. An average of 1200 GDDs were needed to complete their cycle. Different GDDs accumulations were recorded among the genotypes in the two growing seasons due to diverse minimum and maximum temperatures. In particular, over the two years, the CTI 13 accession accumulated the highest GDDs, whilst Montola 2000 and the CTI 5 and CTI 10 genotypes accumulated the lowest GDDs.

### 2.3. Morphological and Yield Components

The year had a significant effect on plant height but it did not determine any significant variation for other parameters. In contrast, morphological and yield components were significantly affected by the genotype. Results of ANOVA revealed that the interaction between the main factors was significant for all morphological and yield parameters in the study, except for the number of branches per plant (Table 2).

Plant height ranged from 113 (first year) to 120 cm (second year); plants, therefore, were taller in the second growing season than in the first season. The safflower genotypes showed high variability for plant height and recorded an average value of 116 cm. In the two years, CTI 17 obtained the highest plant height (138 cm) while CTI 10 produced the lowest height (101 cm), on average. Genotypes produced significant differences in the number of branches per plant. The average value of the number of branches per plant was 13 with a difference of 4 between the highest (14) and the lowest values of this trait (11). The number of capitula per plant was on average 13 and resulted in being significantly affected by genotype. CTI 17 and CTI 5 obtained the highest (15) and lowest (11) average values, respectively, for this morphological trait. Across all the genotypes, TSW ranged between 38 and 41 g with an average value of 39 g. CTI 17 produced the highest TSW in both the growing seasons.

With regard to yield parameters, significant differences between the genotypes were found for seed and oil yield. CTI 17 produced the highest seed (1.40 t ha^−1^) and oil yields (0.58 t ha^−1^), while CTI 13 gave the lowest values for both these yield traits. The average seed and oil yield values for the genotypes were 1.2 t ha^−1^ and 0.46 t ha^−1^, respectively. Seed oil content was found on 40.2% of DM on average and was observed to be higher than 40.0% in four genotypes. It ranged between 38.9% (CTI 11) and 41.3% (CTI 15).

When comparing the productive performance of the safflower genotypes with those of Montola 2000, it was observed that some genotypes performed better than the variety for all yield parameters.

In particular, all the components of crop production were significantly affected by year-by-genotype interactions. The values of these interactions were very similar between the growing seasons (Appendix A). The year-by-genotype interaction showed that the highest number of capitula was determined in the second growing season (CTI 17). Concerning TSW, it was found to be highest in the second growing season, on average, compared to the first growing season. The seed produced by CTI 17 in the second growing season was the heaviest. However, the safflower genotypes obtained similar performance in seed yield in both growing seasons. When observing the year-by-genotype interaction concerning oil content, the highest oil content was found in CTI 15 in the first growing season. The oil yield followed the same trend observed for seed yield and oil content. In the first growing season, CTI 10 showed the best oil yield.

### 2.4. Fatty Acid Composition

Regarding the influence of the main factors on fatty acid composition, it was found that the year had a significant effect only on those fatty acid components which were less abundant. The genotype significantly affected all tested parameters. Results of ANOVA indicated that the year-by-genotype interaction determined significant differences in oil content and stearic acid (Table 3).

Unsaturated fatty acids mainly concerned oleic and linoleic acids and were found to have an average content of 66.1% and 25.1%, respectively, in the two growing seasons. Oleic acid content, in particular, varied among the genotypes from 74.2% (CTI 15) to 56.9% (CTI 10). The highest oleic acid content (74.5%) was obtained by the CTI 15 accession in the 2014–2015 growing season and was greatly genotype dependent. In general, the oleic and linoleic acid contents did not significantly vary over the two years (Table 4).

In the study period, saturated fatty acids mainly concerned palmitic and stearic acids, which were recorded with an average content of 6.14% and 2.04%, respectively. CTI 15 recorded the highest content of palmitic acid in both growing seasons. The level of palmitic acid was found to be higher than that of the stearic acid, on average. The fraction of SFA did not significantly change over the two years (Table 4).

### 2.5. Correlation and Linear Regression Analyses

Correlation analysis (Table 5) showed that seed yield significantly correlated with the number of branches per plant (r = 0.43), TSW (r = 0.40) and oil yield (r = 0.98). No significant differences were found for plant height, number of capitula or seed oil content. In addition, the oil yield positively correlated with plant height (r = 0.35), number of branches per plant (r = 0.42), and TSW (r = 0.37). An increase in safflower oil yield was related to an increase in seed yield and other morphological components. No significant correlation was found between oil yield and oil content.

Linear regression analysis (Table 6) showed that seed and oil yields significantly increased with increases in morphological and productive parameters. Seed yield linearly increased with plant height (R^2^ = 9.4), number of branches per plant (R^2^ = 16.1), number of capitula per plant (R^2^ = 6.3), TSW (R^2^ = 14.7) and seed oil yield (R^2^ = 96.9). No significant linear regression was recorded between seed yield and oil content. Regarding oil yield, this parameter linearly increased with plant height (R^2^ = 10.9), number of branches per plant (R^2^ = 17.2) and TSW (R^2^ = 12.2). No significant linear regression was found with the number of capitula and oil content.

### 2.6. Carbon, Hydrogen and Nitrogen Removal by Seed and Crop Residues

When analysing the C-H-N removal by seed and above- and belowground plant parts, the year did not determine significant differences. The genotype significantly affected the nutrient contents in the various plant parts. The year-by-genotype interaction did not significantly influence any parameters (Table 7).

In general, it was observed that safflower genotypes removed a greater amount of C compared to H and N. Furthermore, singular plant parts contributed differently to nutrient removal and this was due to the effect of the genotype, only. Comparing the genotypes, the main removal of C, H and N was found in the seed with respect to above- and belowground plant parts. Particularly, most genotypes performed better than Montola 2000 in terms of C-H-N removal in the seed. Regarding crude protein content, CTI 10 had the highest value (19.8%) while the lowest value (14.9%) was observed in CTI 11 (13.0%). The average crude protein content for all the genotypes was 16.2% DM.

With regard to qualitative characteristics of the biomass, above- and belowground plant parts showed the highest content for C and the lowest for N as a percentage of dry matter (Table 7). The C-H-N fractions were the same in relative proportions for both the above- and belowground biomass in all the genotypes. The year-by-genotype interaction significantly affected the N removed by seed and crop residues. Concerning C and H content, the interaction between the main factors determined significant differences only for C content in the seed.

### 2.7. Aboveground and Belowground Biomass Production

In Figure 2, the results of above/belowground biomass production of the nine safflower genotypes are shown. Mean ± standard deviation values highlighted differences between the above and belowground parts of the genotypes.

The average dry matter production for the aboveground parts for the two growing seasons was 3.23 ± 0.21 t ha^−1^/year. In particular, relatively homogeneous aboveground production values were recorded across the genotypes ranging between 2.65 (CTI 1) and 3.83 (CTI 13) t ha^−1^. Only five genotypes obtained aboveground biomass values which were higher than 3.0 t ha^−1^. Regarding average dry matter production for the belowground parts, this was found to be 0.80 ± 0.11 t ha^−1^/year. The belowground biomass was 25% of the aboveground biomass. The highest value (0.97 t ha^−1^) was obtained by CTI 13, whilst the lowest value (0.72 t ha^−1^) was recorded in CTI 10 and CTI 17. None of the genotypes obtained average belowground biomass values higher than 1.0 t ha^−1^. No relevant differences regarding above- and belowground biomass values among the safflower genotypes were found in the two growing seasons.

### 2.8. Hierarchically Clustered Heat Map Analysis

Heat map analysis enabled the simultaneous visualization of cluster samples (genotypes) and variables (oil content, oil yield and fatty acid compositions) (Figure 3).

Analysis revealed a pair of dendrograms: the first, structured at the top (dendrogram 1) of the heat map, includes the safflower genotypes, and the second on the left (dendrogram 2) shows the variables which affected this distribution.

Dendrogram 1 displays two macro-clusters. On the left side, the cluster includes genotypes CTI 15, CTI 5 and CTI 13, while, on the right side, the cluster encompasses genotypes CTI 9, CTI 10, CTI 1, CTI 6, CTI 11 and CTI 17 genotypes. Regarding the cluster on the left side of dendrogram 1, CTI 15 is clearly separated from genotypes CTI 5 and CTI 13 due to higher palmitic and linoleic acid content and oil yield. On the contrary, CTI 5 and CTI 13 show the highest oleic acid content with respect to CTI 15.

Concerning the cluster on the right side of dendrogram 1, two sub-clusters are documented. The sub-cluster on the left side includes genotypes CTI 9 and CTI 10. These genotypes are clearly separated from the others due to higher oil content, stearic acid content and content of other compounds. Within the sub-cluster on the right side, two groups are identified: the first includes CTI 1 and CTI 6 while the second encloses genotypes CTI 11 and CTI 17.

Observing the heat map, genotypes CTI 1 and CTI 6 are evidently separated from CTI 11 and CTI 17 due to higher oleic and stearic acid content. On the contrary, CTI 11 and CTI 17 exhibit the highest oil yields.

The clusters in dendrogram 2 clearly highlight the main variables which appear to be characteristic of each sample cluster.

## 3. Discussion

In the Mediterranean region, crop production is greatly impacted by climate change, more frequent and intense extreme climate events, together with land degradation and the acidification and salinization of soils [37]. Extremes, such as abnormal maximum temperatures, heat stress, prolonged periods of drought and sudden floods, can cause crop yield losses and crop quality reduction [38]. In this area, crop yield reductions are predicted for in coming years in most agricultural areas and for most crops such as winter cereals, which are generally produced in monoculture. On this basis, cropping system diversification can represent an excellent solution for farmers to reduce the impact of climate change on crop yields with positive effects on agricultural biodiversity and ecosystem services due to better soil conservation and efficient water management [39]. Furthermore, diversification limits biotic and abiotic stresses for crops, improves farmer income and promotes models of agricultural sustainability [14]. As a consequence, one of the biggest challenges for farmers is to find low-input winter crops to rotate with fall/winter cereals in order to diversify the cropping systems at the arm level. Underutilized oilseed crops, such as safflower, provide a series of agronomic benefits, they are capable of growing in marginal environments and they do not require high energy inputs [40]. In keeping with this scenario, in the present study, eight high oleic safflower genotypes of different origins and a commercial variety used as a reference were tested for the first time over two growing seasons under rainfed conditions in a Sicilian area where cereal crops, such as durum wheat (*Triticum durum* Desf.) and barley (*Hordeum vulgare* L.), are commonly cultivated. The main aim of the study was to demonstrate the effect of genotype × year factor on safflower yield parameters.

All genotypes adequately adapted to the soil and climate condition of the study area and showed good performances in terms of seed and biomass yield, as well as oil content and yield. The growth cycle was completed in 189 and 196 days in the first and second growing seasons, respectively, in accordance with observations from previous studies carried out in other Italian regions [7,14,25]. Although the cycle length of the safflower genotypes was quite different over the two growing seasons, the GDD accumulated from germination to the fruit ripening stage was not affected by the environment and genotype. In both years, significant differences in terms of morphological and yield parameters were recorded among the safflower genotypes and this confirmed the genotype × year effect. Some authors have explained in detail the influence of genetic and environmental factors on safflower yield. For example, Alizadeh and Carapetian [41] stated that genetic variation in safflower germplasm grown in rainfed cold drylands can greatly influence seed yield. Koutroubas et al. [42] affirmed that soil and climate factors can significantly affect the yield performances of safflower. In the present study, the seed and oil yield performances were due to genotype response to climate conditions. Regarding seed yield, the literature shows average values ranging from 1.0 to 5.0 t ha^−1^ depending on genotype, cultivation practices and environmental conditions [25,43,44,45]. In our study, the average seed yield (1.14 t ha^−1^) of the eight genotypes fell within this range but was lower than those found in other Italian areas. For example, in North Central Italy, Zanetti et al. [14] carried out a multi-year and multi-location study, over multiple growing seasons, on a commercial safflower high oleic variety and found an average seed yield of 1775 kg DM ha^−1^. In Central Italy, Abou Chehade et al. [7] evaluated the effects of the genotype and growing season on crop phenology, morphological and yield components over two growing seasons and obtained an average seed yield of 1.93 Mg DW ha^−1^. These differences can be explained by considering the effect of different environmental conditions on genotype performance and, obviously, how agronomic factors, such as the sowing date, plant density, fertilization, and irrigation affect seed yield. In the present study, the fact that no significant differences were found between the two growing seasons was mainly due to similar climate conditions in the two years. No relevant differences in terms of rainfall rates and average temperature values were observed during the vegetative and reproductive phases in each growing season. Several studies [45,46] highlight that dry conditions during the growing season negatively affect the agronomic performance of safflower. It is well known that safflower is tolerant to drought and heat and that the flowering stage is the most sensitive to environmental stress [7]. In this study, safflower was sown in the fall while the flowering stage occurred in April/May before the air temperature significantly increased. This avoided abiotic stress conditions for the plants and facilitated subsequently seed production in both years. As stated by Abou Chehade et al. [7], one of the most important traits of seed quality is protein content due to its use as livestock feed. It is well known that seed protein content in safflower ranges between 10 and 22%; this percentage was also confirmed by the present study. It is worth noting that seed protein content was significantly affected by all the main factors and year-by-genotype interaction. Therefore, it is possible to affirm that different seed protein levels can be expected in response to climate conditions, genotype and their interaction in accordance with the literature.

Concerning morphological characteristics and yield components, evaluation of their relationship is required in order to understand how the components of the safflower yield are related to morphological characteristics. In the present study, the number of branches per plant, the number of capitula per plant and TSW were significantly correlated to seed yield. This was also confirmed by other studies which stated that the number of capitula per plant was the most positively correlated morphological characteristic to safflower yield [25,47,48]. Despite many studies reporting a high correlation between oil content and oil yield, no significant relationship was found between these parameters in the present study. The main reason was the fact that a high number of safflower genotypes had low seed yield, which negatively affected the oil yield value, calculated by the product of seed oil content and seed yield, and, consequently, the level of correlation with the oil content. Most safflower genotypes had similar TSW, which was not significantly affected by the growing season. It is worth noting that plant height was the only morphological characteristic to be affected by genotype and year factors, and year-by-genotype interaction, confirming that this trait is controlled by both genetic and environmental factors [44,49].

Crop residue yield was significantly affected by genotype and year-by-genotype interaction. In particular, the aboveground plant parts removed significant amounts of C and H, while N content was very low due to prolonged soil drought during the two growing seasons. The average N content was, in fact, around 1.0% DM, which allowed us to obtain production values of 35 kg N ha^−1^ in some safflower genotypes. These data can be viewed as positive if the aboveground parts can return to the soil once they are incorporated with tillage, thereby promoting an increase in organic matter rates and improving soil fertility. In the case of use for energy production, however, N content in the aboveground parts would be assessed as negative due to possible NO_x_ release during the combustion process. Regarding belowground plant parts, the biomass yield was also affected by climate conditions during the test period and was equal to 25% of the aboveground biomass, on average. The N content was on average lower than that of aboveground parts which led to low N ha^−1^ production. The complex and difficult removal of root residues from soil suggests leaving these residues in the soil, thereby promoting the transformation of a part of them into stable humus in accordance with sustainable agriculture criteria.

Various authors [14,36,50] affirm that seed oil content and oil yield are the main traits for assessing safflower genotypes over various growing seasons and for introducing this crop into new cropping systems or agricultural areas. However, the seed oil content can largely vary depending on genetic and environmental factors and agronomic practices [24,51,52]. Anjani and Yadav [21], for example, noted that a decrease of 3–5% in oleic acid content is expected when safflower is grown in warmer climates under rainfed conditions. Seghal et al. [53] confirmed this concept and stated that under drought conditions the decrease in oil content is due to a reduction in the concentration of digestible carbohydrates and the unloading of sugars from stem to seeds. Hamdan et al. [54] affirm that in safflower genotypes the high oleic content represents a characteristic which can be considered environmentally stable and genetically controlled. In agreement with this statement, in the present study, no significant differences were recorded between the two growing seasons; moreover, the oleic acid content was found to be stable in each safflower genotype. This means that, in the same environment, despite small changes in temperature trends and rainfall levels, the effect of climate conditions does not determine relevant variations in the oleic acid content of high oleic acid safflower genotypes. In contrast, as demonstrated by Zanetti et al. [14], when comparing two different growing regions, it is possible to find significant differences in oleic acid content. In any case, previous studies which investigated the response of fatty acid composition to climate conditions during seed maturation reported conflicting results. Therefore, there is no clear evidence regarding how environmental conditions affect the fatty acid composition and, in particular, the oleic acid content in safflower genotypes. This can represent a serious problem when seeking to determine a quality standard for safflower oil with reference to industrial and food uses.

The literature reports different average values of oil content mainly due to cultivation environments and cropping practices. As highlighted by La Bella et al. [25], an oil content of 26–37% was found in Greece [55], 24–40% in China [56], 23–40% in Iran [57], 26–36% in Egypt [43] and 16–32% in Turkey [58]. In the present study, oil content was significantly affected by the genotype factor and year-by-genotype interaction. The effect of the growing season was not significant due to the fact that temperature trends and rainfall rates were similar in both years. By comparing the present findings with others obtained under rainfed conditions in the Mediterranean area, the average oil content (40.1%) fell within the range (25–42%) recorded in previous studies [7,25,45]. Oil yield followed the same trend as oil content; in particular, an oil yield ranging from 0.20 to 0.70 t ha^−1^, on average, was reported in safflower genotypes grown in arid and semi-arid regions [22,32,45,55], a finding that was also confirmed in the present study.

## 4. Materials and Methods

### 4.1. Experimental Setup and Main Cultivation Practices

A total of 8 high oleic *Carthamus tinctorius* L. genotypes, provided by the Regional Plant Introduction Station Washington State University (WRPIS) of the United States Department of Agriculture (USDA) were used as plant material and tested. All genotypes (code CTI) were spiny and different in origin (Table 8). The genotypes were compared to the Montola 2000 cultivar, which was used as a reference. At the time of the research, the 8 genotypes had never been tested in Italy.

Field experiments were carried out at the “Calogero Amato Vetrano” Agricultural Technical Institute (Sciacca, Italy, 37°30′43″ N, 13°07′32.08″ E; 110 m a.s.l.), in the south-west of Sicily, during the growing seasons of 2014–2015 and 2015–2016. The soil of the experimental area was classified as Regosol by USD, is sandy clay loam soil. According to the Köppen–Geiger climate classification [59], the study location is characterized by a warm temperate climate with dry summer and mild winter.

Experimental plots were arranged in a randomized complete block design [60] (Gomez and Gomez, 1984) with three replications for each growing season. Each plot measured 15 m^2^ (5 × 3 m). The previous crop was *Triticum durum* Desf. Conventional tillage and mineral fertilization were adopted. The soil was ploughed and harrowed beforehand. Sowing occurred on 18 December 2014 and 20 December 2015, respectively. A density of 50 viable seeds m^−2^ was used and row spacing was 50 cm. Before sowing, 80.0 kg ha^−1^ of phosphorus (P) fertilizer was applied. A total of 100.0 kg ha^−1^ of nitrogen (N) fertilizer was used for the tests, 50.0 kg ha^−1^ at sowing time and 50.0 kg ha^−1^ at the start of stem elongation. The genotypes were grown under rainfed conditions in each growing season. Dicotyledonous weeds were mechanically controlled while fluazifop-p-butyl 13:40% was applied at a rate of 1.0 l ha^−1^ for graminaceous weeds whenever needed. Insect control was carried out by dimethoate 98.0% at a rate of 1.50 l ha^−1^ at the beginning of the flowering stage. Harvest was conducted at seed ripening when the seed moisture content was below 8.0%. A combine harvester equipped with a wheat-cutting bar was used at an interval of 10 days between 25 June and 10 July during both growing seasons, based on the genotype.

### 4.2. Climatic Data

A weather station belonging to the Sicilian Agro-Meteorological Information Service [61] was used to assess the effect of a number of climate factors on the safflower growth cycle. It was located 500 m from the experimental field. The station was synchronized with Greenwich Mean Time (GMT) in order to operate using synoptic forecast models. It was equipped with an MTX datalogger (model WST1800) and various sensors: wind speed sensor (model Robinson cup VDI with an optoelectronic transducer), global radiation sensor (model Philipp Schenk—8102 thermopile pyranometer) to measure cumulative direct and diffuse solar irradiance, temperature sensor (MTX) (model TAM platinum PT100 thermal resistance with anti-radiation screen), relative humidity sensor (MTX) (model UAM with capacitive transducer with hygroscopic polymer films and anti-radiation screen), rainfall sensor (MTX) (model PPR with a tipping bucket rain gauge) and leaf wetness sensor (MTX) (model BFO with PCB). This equipment provided data on the main meteorological parameters in the study.

### 4.3. Plant Growth and Measurements

During each growing season, the main growth stages of safflower genotypes were determined according to Flemmer et al. [62]. The germination stage (BBCH scale code = 00–09) was recorded from the sowing of dry seed to the emergence of cotyledons through the soil surface). The stem elongation stage (BBCH scale code = 30–39) was determined from the beginning of stem elongation to more visibly extended internodes. The flowering stage (BBCH scale code = 30–39) was recorded from the beginning to the end of flowering. The fruit ripening stage (BBCH scale code = 81–89) was recorded up to when the capitula were yellow, fully ripened and ready for harvest. The senescence stage (BBCH scale code = 91–97) was recorded from 10% to 100% of foliage production and most of the capitula became yellow (Figure 4).

The accumulated growing degree days (*GDD*s) were also used to describe crop phenology. Daily *GDD*s were calculated for each phenological stage with the equation [63]:(1)$$GDD=\frac{\left(T_{max}+T_{min}\right)}{2}-T_{base}$$
where *T_max_* and *T_min_* are daily maximum and minimum air temperatures and *T_base_* is the base temperature below which development ceases. A value of 10 °C was used as the base temperature for the safflower plant.

Seed yield was determined on a harvest area of 7 m^2^. Plant height, number of branches per plant, number of capitula per plant and 1000-seed weight (TSW) were recorded on a sample of 20 plants per plot. For each accession, a total of three seed samples were analysed to determine the qualitative characteristics. After harvesting, seeds were cleaned, partially dried, ground to 10 mm size and analysed for their main components. The seed moisture content was determined by oven-drying the seed at 40 °C until constant weight and evaluating the difference in weight before and after treatment.

### 4.4. Oil Content, Crude Protein Content, Fatty Acid Composition and Crop Residue Analysis

The residual oil content was determined with an E-816 ECE extraction unit, by the continuous Twisselmann extraction method using hexane as the solvent [64]. Oil yield was calculated by multiplying dry seed yield by oil content.

The total content of carbon, hydrogen and nitrogen (C-H-N) in seed and crop residues was obtained by dry combustion through an elemental analyser LECO CHN TruSpec. The crude protein content was expressed as a percentage of dry matter (DM) and calculated from nitrogen using the conventional factor of 6.25 [65,66].

The fatty acid (FA) composition was determined by extracting the oil from ground seeds by hexane and trans-methylated in 2NKOH methanol solution [67]. FA methyl ester composition was evaluated by gas chromatography equipped with a flame ionization detector (Carlo Erba HRGC 5300 MEGA SERIES) and a capillary column Restek RT × 2330 (30 m × 0.25 mm × 0.2 µm), following the internal normalization method [68].

### 4.5. Statistical Analysis

Statistical analysis was performed using the software MINITAB 19 for Windows. Data were submitted to the analysis of variance (ANOVA). Genotype and year were used as fixed effects in the linear model/ANOVA. When ANOVA revealed statistically different means, the Tukey test was used to separate means (*p* ≤ 0.01). Correlation and linear regression analyses were carried out to evaluate relationships between the morphological and productive parameters of the safflower genotypes. Values of above- and belowground biomass are shown as mean ± standard deviation of calculations. A hierarchically clustered heat map analysis was also conducted on standardized values using Euclidean distances as a measure of (dis)similarity among the samples (genotypes) regarding oil content, oil yield and fatty acids composition and hierarchical clustering with complete linkage. The heat map was obtained using a program package online (https://biit.cs.ut.ee/clustvis/) (accessed date: 3 February 2023).

## 5. Conclusions

This study shows that high oleic *Carthamus tinctorius* L. represents a useful oilseed crop for the southern regions of Italy and can be introduced into traditional cropping systems in rotation with fall/winter cereals, when grown with winter cycle, thereby diversifying crop production. Due to its high tolerance to drought, soil salinity and reduced agricultural inputs needs, safflower proves itself as an adaptable crop able to obtain a satisfactory productive performance in a semi-arid area of Sicily under rainfed conditions. All tested genotypes provided good results in terms of seed and biomass yield, as well as oil content and yield. Over two growing seasons, climate factors did not determine significant differences in morphological and yield parameters while genotype factor and the year-by-genotype interaction produced notable variations. The oleic acid content was found to be stable in each safflower genotype confirming that it can be considered as an environmentally stable and genetically controlled trait. Regarding crop residues, due to carbon, hydrogen and nitrogen removal rates, the findings highlighted that the most promising use could be to incorporate them into the soil with tillage to improve soil fertility in accordance with sustainable agriculture criteria. However, further studies are needed to compare the safflower genotype performance in various Sicilian environments and better understand how potentially different climate conditions can affect the yield and qualitative characteristics of high oleic acid safflower. Greater attention needs to be paid to understanding the effect of the environment on fatty acid composition and, in particular, on the oleic acid content based on the final use of the safflower oil. This appears essential for upcoming large-scale cultivation of this species in Southern Italy.

## Figures and Tables

**Figure 1 plants-12-01733-f001:**
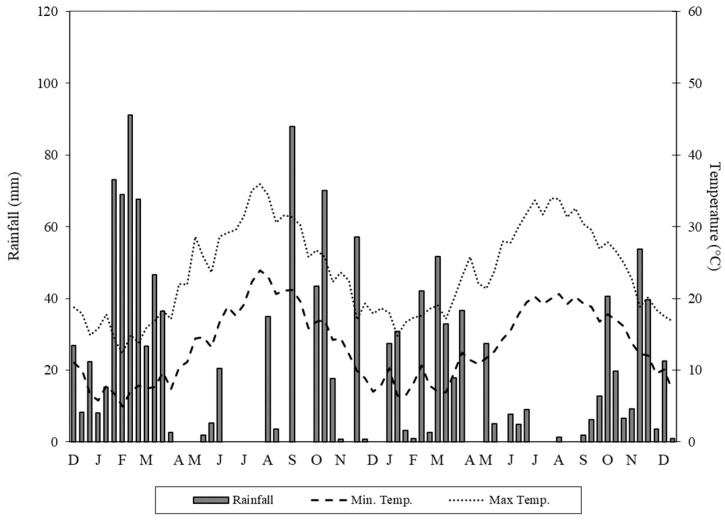
Rainfall and temperature trends during the two growing seasons.

**Figure 2 plants-12-01733-f002:**
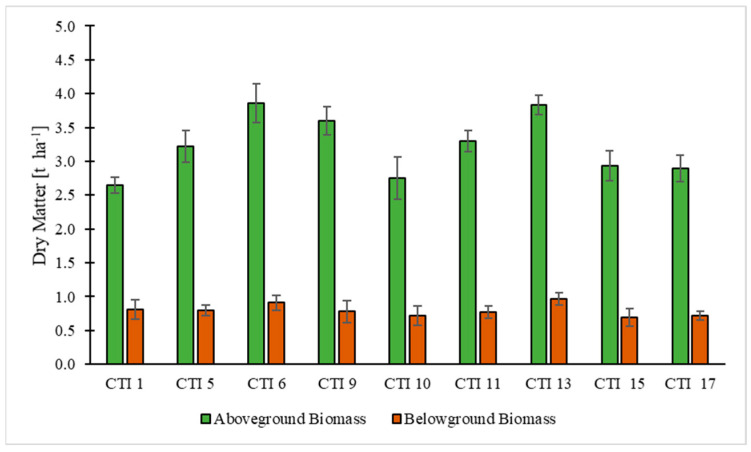
Above- and belowground biomass of the safflower genotypes. Bars indicate the standard deviation of the mean.

**Figure 3 plants-12-01733-f003:**
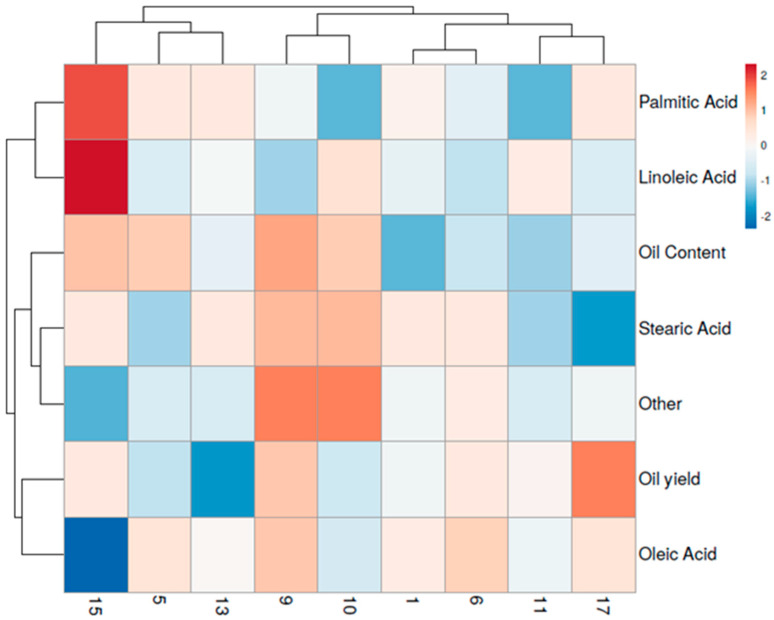
Hierarchically clustered heat map on oil content, oil yield and fatty acid composition of safflower genotypes grown in two growing seasons. Each sample represents a genotype. Data values were transformed to a colour scale.

**Figure 4 plants-12-01733-f004:**
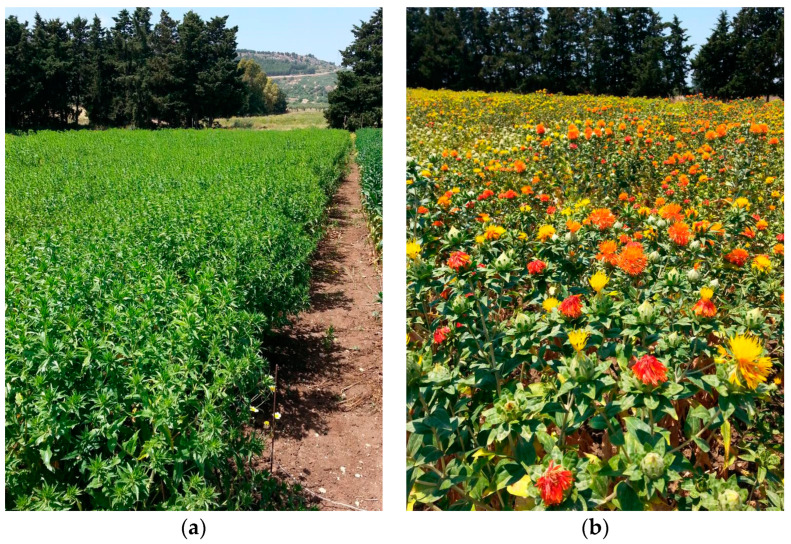
A view of the safflower experimental field in different growth stages: (**a**) stem elongation stage; (**b**) flowering stage.

**Table 1 plants-12-01733-t001:** Duration and cumulative growing degree days (GDD) required from various phenological stages of 9 safflower genotypes across two growing seasons.

Genotype	Duration (Days)					GDD (°C Day)				
	Germination	Stem Elongation	Flowering	Fruit Ripening	Senescence	Germination	Stem Elongation	Flowering	Fruit Ripening	Senescence
2014–2015										
CTI 1	11	66	139	174	188	35	170	538	907	1099
CTI 5	9	62	132	172	186	33	162	463	880	1071
CTI 6	12	73	139	178	191	35	186	538	967	1140
CTI 9	8	65	133	177	191	30	169	470	952	1140
CTI 10	10	64	136	174	186	34	166	501	907	1071
CTI 11 (Montola 2000)	9	61	131	171	183	33	160	455	868	1031
CTI 13	11	72	135	179	192	35	184	489	982	1153
CTI 15	10	62	133	174	187	34	162	470	907	1085
CTI 17	10	65	135	178	189	34	169	489	967	1112
2015–2016										
CT1	14	71	138	179	197	60	274	658	1063	1319
CTI 5	13	74	141	181	193	56	285	678	1089	1262
CTI 6	12	69	137	184	195	52	265	652	1135	1290
CTI 9	14	75	139	182	196	60	287	66	1104	1304
CTI 10	13	73	139	181	193	56	282	665	1089	1262
CTI 11 (Montola 2000)	11	74	140	182	190	49	285	672	1104	1214
CTI 13	12	71	142	186	201	52	274	683	1158	1378
CTI 15	13	73	142	187	197	56	282	683	1172	1319
CTI 17	12	68	136	184	199	52	261	649	1135	1349

**Table 2 plants-12-01733-t002:** Morphological and yield parameters of the 9 safflower genotypes during two consecutive growing seasons.

Main Variables	Plant Height (cm)	Number of Branches (n)	Number of Capitula (n)	TSW (g)	Seed Yield (t ha^−1^)	Oil Content (%)	Oil Yield (t ha^−1^)
Year (Y)							
2014–2015	118 ^a^	12 ^a^	13 ^a^	39 ^a^	1.16 ^a^	40.2 ^a^	0.47 ^a^
2015–2016	113 ^b^	13 ^a^	13 ^a^	39 ^a^	1.13 ^a^	40.1 ^a^	0.45 ^a^
Genotype (G)							
CTI 1	106 ^de^	12 ^bcd^	13 ^abc^	38 ^b^	1.1 ^abc^	38.6 ^c^	0.45 ^abc^
CTI 5	106 ^de^	11 ^d^	11 ^d^	38 ^b^	0.9 ^bc^	41.2 ^ab^	0.39 ^bc^
CTI 6	110 ^d^	13 ^abc^	12 ^bcd^	39 ^ab^	1.2 ^ab^	39.3 ^c^	0.49 ^ab^
CTI 9	120 ^bc^	14 ^abc^	14 ^ab^	38 ^b^	1.3 ^ab^	41.6 ^a^	0.52 ^ab^
CTI 10	101 ^e^	12 ^cd^	11 ^cd^	39 ^ab^	1.0 ^bc^	41.1 ^ab^	0.41 ^bc^
CTI 11 (Montola 2000)	113 ^cd^	13 ^abc^	13 ^bc^	39 ^b^	1.2 ^ab^	38.9 ^c^	0.47 ^abc^
CTI 13	119 ^c^	13 ^abc^	14 ^ab^	38 ^b^	0.8 ^c^	39.8 ^bc^	0.33 ^c^
CTI 15	127 ^b^	14 ^ab^	14 ^ab^	38 ^b^	1.2 ^abc^	41.3 ^a^	0.48 ^ab^
CTI 17	138 ^a^	14 ^a^	15 ^a^	41 ^a^	1.4 ^a^	39.7 ^c^	0.58 ^a^
Source of variation (*p*-Value)							
Y	0.0 **	0.62 ^n.s.^	0.57 ^n.s.^	0.61 ^n.s.^	0.61 ^n.s.^	0.91 ^n.s.^	0.52 ^n.s.^
G	0.0 **	0.0 **	0.0 **	0.002 **	0.0 **	0.0 **	0.0 **
Y × G	0.05 *	0.42 ^n.s.^	0.001 **	0.03 *	0.007 **	0.0 **	0.04 *

Means followed by the same letter are not significantly different for *p* ≤ 0.01 according to Tukey’s test. *, ** significant at the 0.05 and 0.01 probability levels, respectively; n.s. not significant.

**Table 3 plants-12-01733-t003:** Fatty acid composition of the 9 safflower genotypes during two consecutive growing seasons.

Main Variables	Linoleic Acid (%)	Oleic Acid (%)	Palmitic Acid (%)	Stearic Acid (%)	Others (%)
Year (Y)					
2014–2015	24.9 ^a^	66.4 ^a^	6.0 ^a^	2.1 ^a^	0.7 ^a^
2015–2016	25.3 ^a^	66.0 ^a^	6.0 ^a^	2.1 ^a^	0.7 ^a^
Genotype (G)					
CTI 1	26.1 ^c^	65.0 ^cd^	6.2 ^ab^	2.1 ^c^	0.6 ^a^
CTI 5	24.0 ^cde^	67.2 ^bc^	6.3 ^a^	1.9 ^d^	0.5 ^a^
CTI 6	21.1 ^def^	70.1 ^b^	6.0 ^bc^	2.1 ^c^	0.7 ^a^
CTI 9	19.5 ^ef^	71.1 ^ab^	6.1 ^ab^	2.2 ^bc^	1.0 ^a^
CTI 10	34.3 ^a^	56.9 ^e^	5.6 ^d^	2.2 ^ab^	1.0 ^a^
CTI 11 (Montola 2000)	31.5 ^ab^	60.7 ^de^	5.6 ^d^	1.9 ^d^	0.5 ^a^
CTI 13	28.4 ^bc^	62.5 ^d^	6.3 ^a^	2.1 ^bc^	0.5 ^a^
CTI 15	16.6 ^f^	74.2 ^a^	5.8 ^cd^	2.3 ^a^	1.1 ^a^
CTI 17	24.2 ^cd^	67.1 ^bc^	6.3 ^a^	1.8 ^d^	0.6 ^a^
Source of variation (*p*-Value)					
Y	0.60 ^n.s.^	0.50 ^n.s.^	0.55 ^n.s.^	0.70 ^n.s.^	0.67 ^n.s.^
G	0.0 **	0.0 **	0.0 **	0.0 **	0.34 ^n.s.^
Y × G	0.63 ^n.s.^	0.64 ^n.s.^	0.067 ^n.s.^	0.0 **	0.48 ^n.s.^

Means followed by the same letter are not significantly different for *p* ≤ 0.01 according to Tukey’s test. ** significant at 0.01 probability level, respectively; n.s. not significant.

**Table 4 plants-12-01733-t004:** Oil content and composition of saturated and unsaturated fatty acids in 9 safflower genotypes during two consecutive growing seasons.

	Year	2014–2015								
	Genotype	CTI 1	CTI 5	CTI 6	CTI 9	CTI 10	CTI 11	CTI 13	CTI 15	CTI 17
	Oil content (%)	0.49 ^abc^	0.45 ^abc^	0.51 ^ab^	0.56 ^ab^	0.46 ^abc^	0.51 ^ab^	0.27 ^c^	0.46 ^abc^	0.51 ^abc^
Saturated fatty acids (%)	Palmitic acid	6.3 ^a^	6.4 ^a^	6.0 ^a^	6.0 ^a^	5.5 ^a^	5.7 ^a^	6.4 ^a^	5.7 ^a^	6.3 ^a^
	Stearic acid	2.1 ^cde^	1.8 ^f^	2.1 ^cd^	2.3 ^ab^	2.2 ^bc^	1.9 ^ef^	2.1 ^cde^	2.2 ^bc^	1.9 ^def^
	Total SFA	8.4 ^a^	8.3 ^ab^	7.7 ^cd^	7.9 ^bcd^	7.6 ^d^	7.5 ^d^	8.4 ^a^	8.4 ^a^	7.9 ^bcd^
Unsaturated fatty acids (%)	Oleic acid	65.3 ^a^	69.4 ^a^	69.4 ^a^	71.0 ^a^	56.7 ^a^	60.6 ^a^	62.5 ^a^	74.5 ^a^	67.1 ^a^
	Linoleic acid	25.8 ^a^	21.8 ^a^	21.9 ^a^	19.8 ^a^	34.5 ^a^	31.6 ^a^	28.2 ^a^	16.5 ^a^	24.2 ^a^
	Total UFA	91.1 ^a^	91.1 ^a^	91.2 ^a^	90.8 ^a^	91.2 ^a^	92.0 ^a^	90.7 ^a^	91.0 ^a^	91.3 ^a^
	UFA/SFA	10.9 ^a^	11.1 ^a^	11.3 ^a^	10.9 ^a^	11.8 ^a^	12.2 ^a^	10.8 ^a^	11.6 ^a^	11.0 ^a^
	**Year**	**2015–2016**								
	**Genotype**	**CTI 1**	**CTI 5**	**CTI 6**	**CTI 9**	**CTI 10**	**CTI 11**	**CTI 13**	**CTI 15**	**CTI 17**
	Oil content (%)	0.41 ^bc^	0.35 ^bc^	0.47 ^abc^	0.49 ^abc^	0.36 ^bc^	0.44 ^abc^	0.39 ^bc^	0.51 ^ab^	0.65 ^a^
Saturated fatty acids (%)	Palmitic acid	6.2 ^a^	6.2 ^a^	6.1 ^a^	6.2 ^a^	5.6 ^a^	5.6 ^a^	6.2 ^a^	6.0 ^a^	6.4 ^a^
	Stearic acid	2.1 ^cd^	1.9 ^def^	2.1 ^cd^	2.1 ^cde^	2.2 ^abc^	1.9 ^ef^	2.2 ^bc^	2.4 ^a^	1.8 ^f^
	Total SFA	8.4 ^a^	8.3 ^ab^	8.1 ^abc^	8.2 ^ab^	8.1 ^abc^	8.1 ^abc^	8.2 ^abc^	8.3 ^ab^	8.3 ^ab^
Unsaturated fatty acids (%)	Oleic acid	64.6 ^a^	65.1 ^a^	70.8 ^a^	71.3 ^a^	57.0 ^a^	60.7 ^a^	62.4 ^a^	73.8 ^a^	67.0 ^a^
	Linoleic acid	26.5 ^a^	26.2 ^a^	20.2 ^a^	19.3 ^a^	34.1 ^a^	31.4 ^a^	28.7 ^a^	16.7 ^a^	24.1 ^a^
	Total UFA	91.1 ^a^	91.3 ^a^	91.0 ^a^	90.5 ^a^	91.1 ^a^	91.9 ^a^	91.1 ^a^	90.5 ^a^	91.1 ^a^
	UFA/SFA	11.0 ^a^	11.2 ^a^	11.1 ^a^	10.9 ^a^	11.6 ^a^	12.3 ^a^	10.8 ^a^	10.8 ^a^	11.2 ^a^

Means followed by the same letter are not significantly different for *p* ≤ 0.01 according to Tukey’s test.

**Table 5 plants-12-01733-t005:** Pearson’s correlation coefficients for morphological and yield parameters of safflower genotypes.

	Plant Height (cm)	No. Branches	No. Capitula	Seed Yield (t ha^−1^)	TSW (g)	Oil Content (%)	Oil Yield (t ha^−1^)
Plant height (cm)	1.00						
No. branches	0.56 **	1.00					
No. capitula	0.61 **	0.60 **	1.00				
Seed yield (t ha^−1^)	0.334	0.43 **	0.284	1.00			
TSW (g)	0.263	0.24	0.295	0.40 **	1.00		
Oil content (%)	0.03	−0.159	−0.226	−0.242	−0.191	1.00	
Oil yield (t ha^−1^)	0.35 **	0.42 **	0.251	0.98 **	0.37 **	−0.074	1.00

** significant at 0.01 probability level.

**Table 6 plants-12-01733-t006:** Regression equation and coefficient of determination (R^2^).

Independence Variable (X)	Year	Dependence Variable (Y)	Regression Equation	R^2^ (%)	*p*-Value
Plant height (cm)	2014–2015	Seed yield	Y = 1.489 − 0.003 X	0.0	0.444
		Oil yield	Y = 0.515 − 0.0004 X	0.0	0.792
	2015–2016	Seed yield	Y = −1.264 + 0.021 X	56.2	0.000
		Oil yield	Y = −0.421 + 0.008 X	53.1	0.000
Number of branches (n)	2014–2015	Seed yield	Y = 0.893 + 0.021 X	0.0	0.513
		Oil yield	Y = 0.304 + 0.013 X	0.0	0.340
	2015–2016	Seed yield	Y = −0.348 + 0.113 X	37.1	0.000
		Oil yield	Y = −0.065 + 0.040 X	31.9	0.001
Number of capitula (n)	2014–2015	Seed yield	Y = 1.633 − 0.036 X	1.2	0.264
		Oil yield	Y = 0.627 − 0.012 X	0.0	0.366
	2015–2016	Seed yield	Y = 0.141 + 0.077 X	25.6	0.004
		Oil yield	Y = 0.113 + 0.026 X	20.8	0.010
TSW (g)	2014–2015	Seed yield	Y = 0.486 + 0.017 X	0.0	0.539
		Oil yield	Y = 0.230 + 0.006 X	0.0	0.604
	2015–2016	Seed yield	Y = −2.981 + 0.105 X	34.2	0.001
		Oil yield	Y = −1.037 + 0.038 X	31.7	0.001
Oil content (%)	2014–2015	Seed yield	Y = 0.552 + 0.015 X	0.0	0.556
		Oil yield	Y = −0.257 + 0.018 X	7.8	0.085
	2015–2016	Seed yield	Y = 6.704 − 0.138 X	35.2	0.001
		Oil yield	Y = 2.238 − 0.044 X	24.7	0.005
Seed yield (t ha^−1^)	2014–2015	Oil yield	Y = −0.004 + 0.406 X	94.7	0.000
	2015–2016	Oil yield	Y = 0.031 + 0.372 X	98.6	0.000
Oil yield (t ha^−1^)	2014–2015	Seed yield	Y = 0.069 + 2.337 X	94.7	0.000
	2015–2016	Seed yield	Y = −0.068 + 2.652 X	98.6	0.000

**Table 7 plants-12-01733-t007:** Carbon, hydrogen and nitrogen content as a percentage of dry matter in seed, above- and belowground crop residues of 9 safflower genotypes during two consecutive growing seasons.

Main Variables	Seed				Aboveground Crop Residues			Belowground Crop Residues		
	C (%)	H (%)	N (%)	Protein (%)	C (%)	H (%)	N (%)	C (%)	H (%)	N (%)
Year (Y)										
2014–2015	63.8 ^a^	8.4 ^a^	2.6 ^a^	16.6 ^a^	48.0 ^a^	6.6 ^a^	0.6 ^a^	47.1 ^a^	6.3 ^a^	0.4 ^a^
2015–2016	64.0 ^a^	8.4 ^a^	2.5 ^b^	15.8 ^b^	47.5 ^a^	6.6 ^a^	0.6 ^a^	46.9 ^a^	6.3 ^a^	0.4 ^a^
Genotype (G)										
CTI 1	64.2 ^abc^	8.5 ^a^	2.5 ^de^	15.7 ^d^	47.7 ^a^	6.7 ^a^	0.6 ^b^	47.7 ^a^	6.4 ^a^	0.3 ^b^
CTI 5	64.4 ^ab^	8.4 ^ab^	2.7 ^bc^	16.7 ^bc^	47.8 ^a^	6.6 ^a^	0.5 ^bcd^	46.6 ^a^	6.3 ^ab^	0.4 ^b^
CTI 6	63.9 ^abc^	8.3 ^ab^	2.7 ^bc^	16.6 ^bc^	47.1 ^a^	6.5 ^a^	0.4 ^d^	47.8 ^a^	6.4 ^a^	0.3 ^b^
CTI 9	64.6 ^a^	8.5 ^ab^	2.5 ^de^	15.6 ^de^	47.1 ^a^	6.5 ^a^	0.5 ^bcd^	47.7 ^a^	6.4 ^a^	0.4 ^b^
CTI 10	63.7 ^bc^	8.6 ^a^	3.2 ^a^	19.8 ^a^	49.3 ^a^	6.5 ^a^	0.9 ^a^	47.5 ^a^	6.1 ^b^	0.7 ^a^
CTI 11 (Montola 2000)	61.9 ^d^	8.4 ^ab^	2.1 ^f^	13.0 ^f^	48.3 ^a^	6.6 ^a^	0.6 ^bc^	42.7 ^a^	5.6 ^c^	0.4 ^b^
CTI 13	64.4 ^ab^	8.4 ^ab^	2.6 ^cd^	16.3 ^cd^	47.3 ^a^	6.5 ^a^	0.5 ^cd^	47.8 ^a^	6.4 ^a^	0.4 ^b^
CTI 15	63.5 ^c^	8.2 ^b^	2.4 ^e^	14.9 ^e^	47.4 ^a^	6.6 ^a^	0.5 ^bcd^	47.4 ^a^	6.4 ^a^	0.3 ^b^
CTI 17	64.5 ^ab^	8.3 ^ab^	2.8 ^b^	17.3 ^b^	48.0 ^a^	6.6 ^a^	0.5 ^bcd^	47.4 ^a^	6.4 ^a^	0.4 ^b^
Source of variation (*p*-Value)										
Y	0.15 ^n.s.^	0.40 ^n.s.^	0.0 **	0.0 **	0.13 ^n.s.^	0.44 ^n.s.^	0.63 ^n.s.^	0.31 ^n.s.^	0.75 ^n.s.^	0.21 ^n.s.^
G	0.0 **	0.005 **	0.0 **	0.0 **	0.06 ^n.s.^	0.18 ^n.s.^	0.0 **	0.06 ^n.s.^	0.0 **	0.0 **
Y × G	0.0 **	0.58 ^n.s.^	0.0 **	0.0 **	0.16 ^n.s.^	0.06 ^n.s.^	0.0 **	0.14 ^n.s.^	0.38 ^n.s.^	0.0 **

Means followed by the same letter are not significantly different for *p* ≤ 0.01 according to Tukey’s test. ** significant at 0.01 probability level, respectively; n.s. not significant.

**Table 8 plants-12-01733-t008:** The information on safflower genotypes grown under rainfed conditions during the 2014–2015 and 2015–2016 cropping seasons.

Accession/Code	Spiny/Spineless	Pollinated Type	Linoleic/Oleic Type	Origin
CTI 1	spiny	open-pollinated	oleic	Bangladesh
CTI 5	spiny	open-pollinated	oleic	Bangladesh
CTI 6	spiny	open-pollinated	oleic	USA
CTI 9	spiny	open-pollinated	oleic	USA
CTI 10	spiny	open-pollinated	oleic	Bangladesh
CTI 11 (Montola 2000)	spiny	open-pollinated	oleic	Italy
CTI 13	spiny	open-pollinated	oleic	USA
CTI 15	spiny	open-pollinated	oleic	India
CTI 17	spiny	open-pollinated	oleic	Italy

## Data Availability

The data presented in this study are available on request from the corresponding author.

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
