# Peer review of "Effects of Genotype and Climate on Productive Performance of High Oleic Carthamus tinctorius L. under Rainfed Conditions in a Semi-Arid Environment of Sicily (Italy)"

_plants, 2023, doi:10.3390/plants12091733_

Round 1

Reviewer 1 Report

Dear Authors,

Author Response

Dear reviewer,

please you will find in attachment the main responses to your comments.

Best regards

Nicolò Iacuzzi on behalf of the authors

Reviewer 2 Report

The manuscript is based on an interesting study to promote the adoption of safflower for crop diversification. The manuscript is very well written. However, addressing the below comments can further improve the quality to some extent.

1. Authors describe that the first year has relatively better and uniform rainfall over the growing season than the second year. Regarding temperature, average minimum and maximum temperatures in the two growing seasons were similar. However, the growth cycle was short in the first season. What could be the possible reason for this? In general, stress leads to early maturity in cultivars.

2. Authors should mention that genotype and year were used as fixed effects in the linear model/ANOVA

3. Only plant height performance has been described for two years, but other traits also showed significant Genotype-Env interaction. Emphasis should also be more on seed and oil yield, as economically important traits. It would be better to put the mean values of such traits year-wise in the table.

4. In linear regression, it seems that the data of two years has been averaged but it should be done separately for traits affected by Genotype-Environment interaction to avoid biases. 

5. Line #285-286.  "On the contrary, CTI 285 5 and CTI 13 showed the highest oleic acid content with respect to CTI 13".   Why respect to CTI 13, should it not be CTI 15? Kindly check.

Author Response

(The authors gave the same response as above.)

Reviewer 3 Report

This manuscript contains useful and interesting information on the effects of genotype and climate on safflower in Italy and should be acceptable after minor revisions.  My comments are:

1.       On line 66, change “On the contrary, 66 oil which is rich in monounsaturated oleic acid shows high oxidative stability, making it 67 suitable to cooking and an alternative to olive oil in arid and semi-arid regions of the 68 world [22].” To “In contrast, . . .”

2.       On line 87:  change “As regards the South of Italy, very limited research on safflower varieties has been 87 carried out in recent years [25] for a variety of reasons, however, mainly due to reluctance 88 of farmers to include minor or underutilised crops in rotation with fall/winter cereals or 89 annual legumes, to the absence of local market, the unavailability of locally adapted vari- 90 eties and to a greater importance given to other oilseed crops.” To “In the  South of Italy, very limited research on safflower varieties has been carried out recently [25] for a variety of reasons.  Farmers are reluctant to include minor or underutilised crops in rotation with fall/winter cereals or  annual legumes.  There is no local market and locally adapted varieties are not available.  There has been greater importance given to other oilseed crops.”

3.        On line 108:  “Despite this event, plants and soil were not damaged.”  I am not sure what this means.  Reword to make clearer.

4.       On line 125:  Change “In the two years of study, the cycle length of the safflower genotypes differed (Table 125 1).” To something like “In the two years of study, the lengths of the growing season differed among the safflower genotypes (Table 125 1).”

5.       Table 1, right five columns:  Round values to nearest whole numbers.

6.       Lines 156-171:  Round these values to nearest whole numbers.

7.       Table 2 first four columns: Round values to nearest whole numbers.

8.       Table 6: The regression equations are far too precise.  Round some of these values off to make more readable.

Author Response

(The authors gave the same response as above.)

Round 2

Reviewer 1 Report

Dear Authors,
The manuscript has been well revised.

Thank you and best regards

Reviewer 2 Report

All queries have been considerably addressed by the authors.